# Cost-Effectiveness Analysis of Green–Gray Stormwater Control Measures for Non-Point Source Pollution

**DOI:** 10.3390/ijerph17030998

**Published:** 2020-02-05

**Authors:** Shi Qiu, Haiwei Yin, Jinling Deng, Muhan Li

**Affiliations:** 1School of Architecture and Urban Planning, Nanjing University, Nanjing 210000, China; qiushinju@163.com (S.Q.); Limuhande163@163.com (M.L.); 2International Institute for Earth System Science, Nanjing University, Nanjing 210000, China; gamlingdang@163.com

**Keywords:** rainfall-runoff, SUSTAIN model, cost-effectiveness analysis, urban non-point source pollution, multi-scenarios analysis, high-intensity rainfall

## Abstract

The control of non-point source pollution (NPS) is an essential target in urban stormwater control. Green stormwater control measures (SCMs) have remarkable efficiency for pollution control, but suffer from high maintenance, operation costs and poor performance in high-intensity rainfall events. Taking the Guilin Road subwatershed in Rizhao, China, as a case study, a scheme for coupling gray and green stormwater control measures is proposed, and the gray SCMs are introduced to compensate for the shortcomings of green SCMs. The System for Urban Stormwater Treatment and Analysis Integration (SUSTAIN) model was employed to investigate the cost-effectiveness of three scenarios (green SCMs only, gray SCMs only, and coupled green–gray SCMs). The results show that the optimal solutions for the three scenarios cost USD 1.23, 0.79, and 0.80 million, respectively. The NPS control ability of the coupled green–gray scenario is found to be better than that of the other two scenarios under rainfall events above moderate rain. This study demonstrates that coupled green–gray stormwater control management can not only effectively control costs, but can also provide better pollution control in high-intensity rainfall events, making it an optimal scheme for effective prevention and control of urban non-point source pollution.

## 1. Introduction

Rapid urbanization and intensive human activity have resulted in an increase in impervious surfaces in most cities. Conversion of pervious land to impervious surfaces will lead to increased runoff volume, peak flow, and runoff pollutant loads. Increasing amounts and types of pollutants are flushed from these surfaces and transported to drainage systems and downstream waters with surface runoff during rainfall events, causing severe non-point source or diffuse pollution. To solve this problem, a series of sophisticated stormwater control measures (SCMs), such as sustainable drainage systems (SUDs), water-sensitive urban design (WSUD), and best management practice (BMP) with low-impact development (LID) have been proposed and implemented around the world [1]. China introduced the concept of the sponge city in 2013, and gradually began to promote pilot projects for the construction of sponge cities in 31 cities starting in 2016. In April 2019, the evaluation standard for sponge cities was released, which stipulated the content and methods for the evaluation of sponge city construction.

The construction of sponge cities entails the creation of urban areas that are less likely to have an adverse impact on the natural environment and that preserve pre-development near-natural hydrological conditions in order to delay the peak runoff and reduce the runoff volume and flushing of pollutants into drainage systems. Decentralized site-based green stormwater control measures (green SCMs), such as green roofs, vegetated filter strips, and bioretention cells, are widely used in realistic projects. Studies based on field observations [2] or numeric models [3,4] have revealed the efficiency of green SCMs for controlling urban runoff and water pollution. Furthermore, studies have also suggested that the pollution control effect of green SCMs during rainfall events with a 2 year recurrence interval is excellent. However, green SCMs are less reliable for dealing with medium- and high-recurrence interval rainfall events [4,5]. Gray stormwater control measures (gray SCMs) such as detention tanks and cisterns, which have a relatively higher capacity for storage and retention, possess the ability to compensate for the limitations of green SCMs under rainfall events above moderate rain [6]. Thus, it is necessary to couple green and gray SCMs to build a more robust and efficient system for reaching urban stormwater control targets [7,8,9,10]. Feasibility of the coupled system has been investigated in existing studies, and results show that coupled green–grey SCMs can effectively control stormwater quantity and quality [11,12,13,14].

Despite their ability to control runoff and pollution, a systematical evaluation is needed to evaluate the cost of coupled green–gray SCMs. Cost-effectiveness analyses and system optimization studies have been carried out in recent years to obtain an optimized solution for stormwater management. Life cycle assessment and the cost–benefit ratio [15,16,17,18] are widely used to conduct cost-effectiveness analyses. Hydrology models coupled with algorithms (e.g., the scatter search algorithm [19], simulated annealing algorithm [20], and genetic algorithm [21,22]) have developed rapidly in recent years, owing to their efficiency and reliability. The System for Urban Stormwater Treatment and Analysis Integration (SUSTAIN) model, developed by the United States Environmental Protection Agency (US EPA), provides an optimization module and a post-processing module. The non-dominated sorting genetic algorithm-II (NGSA-II) is used in the optimization module to perform the search for an optimized cost-effective combination of SCMs that meet a user-defined target. Studies have been implemented in SUSTAIN to investigate urban SCMs and their cost-effectiveness [3,23,24,25].

The question of whether coupled green–gray SCMs can compensate for the shortcomings of single green SCMs still needs to be answered. In addition, it is unclear whether these systems are able to achieve higher cost-effectiveness. To answer these questions, this study employed the SUSTAIN model on the ArcGIS platform to assess the cost-effectiveness of coupled green–gray SCMs. The aims of this study were to: (1) create a SUSTAIN model of the study area to examine the ability of coupled green–gray SCMs to control non-point source pollution; (2) compare the cost-effectiveness of three different stormwater control scenarios under the same pollution reduction target; and (3) investigate the ability of coupled green–gray SCMs to control non-point source pollution during high-intensity rainfall events.

## 2. Materials and Methods

### 2.1. Study Area and Data Preparation

A subwatershed within an economic zone in Rizhao, Shandong Province, China, was selected as the case study site. This site covered 138.8 ha and consisted of industrial land, parking lots, and other facilities (Figure 1). The land use types included building (28.42%), green space (37.05%), road (25.84%), and other impervious surface (8.69%). Table 1 lists the land use constituents in the study area. The site was enclosed by natural subwatershed boundaries and roads; therefore, this area can be considered an independent subwatershed. A separate sewer system for runoff had been built; the networks were constructed using reinforced concrete, and the only outlet was located at the northeastern corner of the study area. Owing to the high intensity of industrial and construction activities in the study area, various pollutants accumulate rapidly on the impervious surfaces. Thus, the total amount of pollutants was relatively large, leading to a prominent problem with non-point source pollution during rainfall events. Therefore, it was necessary to establish a cost-effective scheme for pollution control.

All of the relevant data required to build the SUSTAIN model were collected and processed in accordance with the requirements listed in the model’s manual. Four types of data were used: (1) physiographic and hydro-meteorological data, including land use, digital elevation, soil type map, stream network, climate, and precipitation data; (2) hydraulic data, including manholes, drainage networks, and dimensions; (3) runoff flow and water quality data for calibration; and (4) cost data for different stormwater control measures. An automatic weather station (HOBO U30, H21) was installed in a green space near the outlet, and an Ultrasonic Doppler Flowmeter (STARFLOW-6526H) was installed in the outlet pipe to collect simultaneous and continuous weather and hydrology data. Seventeen runoff samples used for calibration were collected on September 19, 2018 at the outlet and tested within 24 h.

### 2.2. Goodness-of-Fit Test

The calibration of significant operating parameters is essential in building a hydrological model in order to achieve accurate response of the catchment. These parameters in the SUSTAIN model were calibrated for runoff volume and quality before SCM scenario evaluations. The Nash–Sutcliffe efficiency (NSE) index [26] and correlation coefficient (R) [27], were used to evaluate the accuracy of the parameters. Calculation methods of the two indices are shown in Equations (1) and (2), respectively.
(1)$$NSE=1-\frac{\sum_{t=1}^{n}{\left(q_{t}^{obs}-q_{t}^{sim}\right)}^{2}}{\sum_{t=1}^{n}{\left(q_{t}^{obs}-\bar{q_{t}^{obs}}\right)}^{2}}$$
(2)$$R=\frac{\sum_{t=1}^{n}\left(q_{t}^{obs}-\bar{q_{t}^{obs}}\right)\left(q_{t}^{sim}-\bar{q_{t}^{sim}}\right)}{\sqrt{\sum_{t=1}^{n}\left(q_{t}^{obs}-\bar{q_{t}^{obs}}\right)}\sqrt{\sum_{t=1}^{n}\left(q_{t}^{sim}-\bar{q_{t}^{sim}}\right)}}$$
where $q_{t}^{obs}$ is the observed outflow (m^3^/s) or total suspended solids (TSS) concentration (mg/L) at time *t*; $q_{t}^{sim}$ is the model simulated outflow or TSS concentration; $\bar{q_{t}^{obs}}$ is the average observed value and $\bar{q_{t}^{sim}}$ is the simulated value; *t* is time; and *n* is the total number of time steps.

NSE indicates the goodness-of-fit based on a comparison between the simulated and observed outflow and TSS concentration at the outlet. R measures how well the proportion of the total variation of the outcomes is explained by the simulation. The parameters were calibrated by maximizing the two indices for the simulated and observed results at the outlet.

### 2.3. Siting of Stormwater Control Measures

Considering there is no definitive classification between green and gray SCMs, based on the study conducted by Bell et al. [28], SCMs constructed from traditional construction materials such as concrete and stone were considered gray SCMs in this study. While those constructed and filled with various media for storage, filtration, and support of the growth of plants were considered green SCMs.

First, three green SCMs (green roofs, bioretention cells, and vegetated filter strips) and three gray SCMs (infiltration trenches, porous pavement, and cisterns), were selected for this study based on the characteristics of the study area and their stormwater control effects demonstrated in previous studies [29,30,31,32]. These stormwater control measures can effectively reduce runoff volume and pollutant load through infiltration, precipitation, plant uptake, and other mechanisms [25]. Then, the BMP Siting Tool in the SUSTAIN model was employed to site suitable areas for specific SCMs based on the site location, soil conditions, groundwater level, terrain, and other requirements.

### 2.4. Scenario Designs

Four scenarios were designed, simulated, and discussed. The status quo scenario (S0) represents the current post-development situation without green or gray SCMs. The green-SCMs-only scenario (S1) and the gray-SCMs-only scenario (S2) represent situations in which only green SCMs or only gray SCMs are installed in the site, respectively. The coupled green–gray SCMs scenario (S3) represents the situation in which coupled green–gray SCMs are installed at the site, where the SCMs are connected successively in the stormwater treatment train.

The status quo scenario in the SUSTAIN model (S0) was established based on the land use, elevation, and drainage system design. The sub-catchments in the study area were primarily divided using the Hydrology Tool in the ArcGIS Toolbox and then modified based on the Rizhao drainage system plan and topography. The drainage system was generalized into 57 sub-catchments, 54 conduits, 54 manhole junctions, and 1 outlet. Then, precipitation and weather data were entered in time series required by the SUSTAIN model. The Horton equation was used to simulate the infiltration process, the dynamic wave method was used for the hydraulic calculations, and an exponential function was used for the pollutant buildup and wash-off model.

The design parameters for different SCMs were defined in the model based on the manual and “Sponge city Construction Technical Guidelines” [33]. In this study, only the design and construction costs were considered and obtained from relevant realistic projects in Jiangsu, Beijing Province, and data from the International Stormwater BMP Database [34]. Table 2 lists the dimensions and costs of the SCMs.

### 2.5. Optimization and Cost-Effectiveness Analysis

The BMP Optimization module in the SUSTAIN model was used for the cost-effectiveness optimization process; this module implements the NGSA-II algorithm to determine cost-effective solutions. To generate cost-effective solutions for each scenario, SUSTAIN requires the user to specify decision variables including the SCMs installed and a single optimization target. Considering that studies have found there is a high correlation between TSS and total phosphorous (TP), chemical oxygen demand (COD), and other pollutant indices under non-point source pollution in industrial zones [35,36], a 70% annual TSS load reduction was defined as the management target in this study. The SCM configuration parameters were set as the decision variables, the range of each SCM parameter was defined from zero to the maximum, and the search interval was approximately one-tenth of the range. The solution that met the control target with the minimum cost in cost-effectiveness solutions was chosen as the optimal solution; the effectiveness of the solutions for non-point source pollution control under different rainfall event types will be further investigated in the Section 3.3.

## 3. Results

### 3.1. Parameters Calibration

The calibration of hydrological and hydraulic parameters was based on a comparison between the simulated and observed outflow and total suspended solids (TSS) concentration at the outlet. The parameters were first set according to the relevant research results and the model manual. The monitored runoff flow and quality data of one rainfall-runoff event (19 September 2018) was used for the calibration. Figure 2 shows the comparison between observed and simulated outflow hydrograph and TSS concentration for the calibration. The parameter calibration results were determined to be acceptable and are listed in Table 3 and Table 4. The NSE index and R were 0.92 and 0.85, respectively, for the outflow calibration, and 0.90, and 0.77, respectively, for the water quality calibration.

### 3.2. SCM Siting and Scenario Designs

After applying the BMP Siting Tool, suitability maps for the SCMs were obtained, as shown in Figure 3; Figure 3a,b show the suitability maps for the gray and green SCMs, respectively. Bioretention cells were placed in green spaces around the buildings and along the sidewalks. Vegetated filter strips were arranged along the sidewalks of the main roads. Green roofs were placed on the buildings with flat tops. Infiltration trenches were placed along the parking lots, green spaces, and low-traffic roads. Cisterns were installed under the open spaces around buildings. Porous pavement was placed along sidewalks and in parking lots with relatively low traffic loads. The total area of each SCM was calculated, and the results are listed in Table 5. Corresponding to the treatment train and taking suitability into consideration, three SCM scenario schemes were drafted (Figure 4b–d).

### 3.3. Cost-Effectiveness Analysis

A 70% reduction in the annual TSS load was defined as the optimization target, and the cost-effectiveness analysis results are shown in Figure 5. The x-axis denotes the cost, and the y-axis indicates the annual reduction percentage in TSS load relative to the status quo scenario. The data points in Figure 5 represent the non-point source pollution control effectiveness and corresponding cost of each possible solution in a given scenario. The orange points indicate cost-effective solutions, whereas the green point represents the optimal solution in the specific scenario.

For the solutions that achieved the 70% optimization target, the mean costs under scenarios S1, S2, and S3 were USD 1.30, 1.21, and 0.95 million, respectively; the mean TSS load reductions were 70.6%, 73.84%, and 70.92%, respectively. It can be seen that there was no significant difference between the scenarios in terms of the TSS reduction, but the mean cost of S3 was markedly lower than that of the other two scenarios. In other words, coupled green–gray SCMs can meet the same pollution control target as green or gray SCMs alone, and the cost of coupled green–gray SCMs was comparatively lower.

In addition, when the cost of S1, S2, and S3 reached USD 0.71, 0.71, and 0.54 million (with corresponding TSS load reductions of approximately 65%, 68%, and 62%), the cost-effectiveness curve for each scenario showed diminishing marginal benefits. Specifically, the cost-effectiveness curve for S3 was steeper before the TSS reduction reached 62%, indicating that coupled green–gray SCMs can provide a notable marginal benefit when the reduction rate is less than 62%.

As for the optimal solutions, the costs of S1, S2, and S3 were USD 1.23, 0.79, and 0.80 million, respectively. The composition and specific cost of the optimal solutions are listed in Table 6. The optimal solution for S1 had the highest cost, while S2 and S3 had very similar costs, implying that the coupled green–gray SCMs can not only achieve the same non-point source pollution control target as the single SCMs, but also effectively limit costs. Thus, the adoption of coupled green–gray SCMs can provide greater cost-effectiveness than either green or gray SCMs alone.

### 3.4. Effectiveness of Pollution Control Under Different Event Types

Considering the limitation of green SCMs, it is necessary to perform an assessment of the ability of coupled green–gray SCMs to control pollution under high-intensity rainfall events. Owing to the random duration and intensity of different rainfall events, the pollutant concentration varies significantly during the rainfall process. Therefore, the event mean concentration (EMC) was used as the evaluation index in this study [37]. First, 26 rainfall events in 2018 were classified into five event types, light rain (<9.9 mm), moderate rain (10–24.9 mm), heavy rain (25–49.9 mm), storm rain (50–99.9 mm), and heavy rainstorm (>100 mm) events according to the rainfall depth in 24 h [38]. Then, the EMCs of each optimal solution were compared under different event types; the results are shown in Figure 6 and Figure 7. The mean values of the EMCs for S1, S2, and S3 under all event types were 23.17 mg/L, 20.12 mg/L, and 18.13 mg/L, respectively. Compared to the mean EMC of the status quo scenario under all event types (57.69 mg/L), S1, S2, and S3 provided EMC reductions of 61.51%, 66.38%, and 69.93%, respectively, indicating that all of the scenarios can effectively provide significant pollutant control. Moreover, the mean EMC value for S3 was much lower than those for S1 and S2, exhibiting the best control of non-point source pollution among the three SCM scenarios.

Furthermore, Figure 7 shows that with a continuous increase in event type level, the mean reduction percentage under all SCM scenarios exhibits a generally decreasing trend. In event type level above moderate rain (>25 mm), the mean reduction percentages in scenarios S1, S2, and S3 were 53.85%, 58.29%, and 64.72%, respectively. This demonstrates that the pollution control ability of S3 was still stronger than the other two scenarios, indicating that coupled green–gray SCMs can compensate for the limitations of single green or gray SCMs under above moderate intensity rainfall events.

## 4. Discussion

As urban stormwater management concepts and methods develop, non-point source pollution control is getting more attention. The widely installed green SCMs have shown efficiency and shortcoming in runoff volume and pollution control. Many previous studies have suggested coupling green SCMs and gray SCMs in order to build a robust and resilient urban drainage system. Xu et al. [13] indicated that coupled green and gray infrastructure systems could effectively alleviate urban waterlogging during a ten-year return period of rainfall, restore ecosystem services, and generate economic benefit. Bakhshipour et al. [12] also found that hybrid green–blue–gray infrastructures could economically compete with conventional gray-only pipe networks.

In this study, we chose TSS load reduction as the cost-effectiveness optimization target and provided insight to the control effect under high-intensity rainfall events. The main objective of this study was to examine the cost-effectiveness for non-point source pollution control and answer the question of whether coupled green–gray SCMs can compensate for the shortcomings of single green SCMs. The results indicated that coupled green–gray SCMs could attain a same effect of non-point source pollution control with a lower cost under a same pollution control target. Moreover, the results also indicated that the coupled green–gray SCMs exhibited a higher pollution reduction rate under rainfall events above moderate rain. Therefore, implementing a coupled green–gray SCM system is a more cost-effective way to control urban non-point source pollution. By using SUSTAIN model to optimize stormwater control measures, urban governments could find the most cost-effective proposal based on local characteristics and available funding. Also, coupled green–gray SCMs proves to be a more cost-effective combination in urban stormwater management, which should be promoted in constructions of sponge cities in the future.

Three potential limitations of this study should be given more consideration in future studies. Firstly, the hydrology and water quality data used for the calibration were collected during a single rainfall event (moderate rain), and thus it is necessary to perform more observations and data collections under different rainfall intensities to further calibrate and validate the model parameters for more precise results. Secondly, the cost-effectiveness optimization process can only select one target at a time, leading to limitations in the scenario optimization. Therefore, a multi-objective cost-effectiveness analysis should be performed in the future. Finally, this study only performed cost-effectiveness analyses on the benefits of non-point source pollution control. Studies have shown that stormwater control measures have comprehensive social, economic, ecological, and environmental benefits, especially green SCMs [39,40], and thus future studies should also consider the comprehensive ecological benefits provided by stormwater control measures.

## 5. Conclusions

Various stormwater control measures have been applied in practical projects, but some shortcomings arise with the application of either green or gray stormwater control measures alone. In this study, the cost and non-point source pollution control of coupled green–gray SCMs were analyzed by comparing various scenarios for a site in the Rizhao economic zone using the SUSTAIN model on an ArcGIS platform. The results show that for the management target of a 70% annual reduction in the TSS load, the optimal solution with coupled green–gray SCMs cost USD 0.80 million, while the green- or gray-only SCMs scenarios cost USD 1.23 and 0.79 million, respectively. Meanwhile, the mean reduction in TSS under rainfall event types above moderate rain is 64.72% for the coupled SCM scenarios, while the single SCM scenarios provide reductions of 53.85% and 58.29%. These results indicate that coupled green–gray SCMs can achieve similar non-point source pollution control to that of single-type SCMs, with a comparatively lower cost. In rainfall events stronger than moderate rain, the control effect of coupled grey–green SCMs on the EMC is also significantly higher than that of the single-type SCMs, demonstrating that coupled SCMS can effectively compensate for the limitations of single SCM types for the control of non-point source pollution in high-intensity rainfall events.

## Figures and Tables

**Figure 1 ijerph-17-00998-f001:**
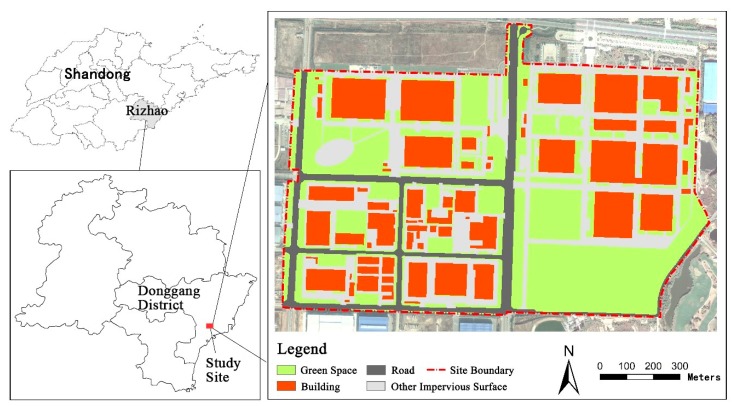
Location of the study site and land use map of the area.

**Figure 2 ijerph-17-00998-f002:**
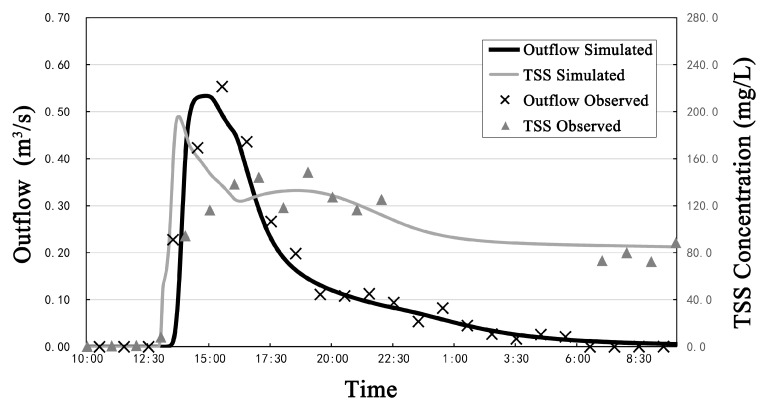
Comparison between simulated and observed results for the outflow and total suspended solids (TSS) concentration.

**Figure 3 ijerph-17-00998-f003:**
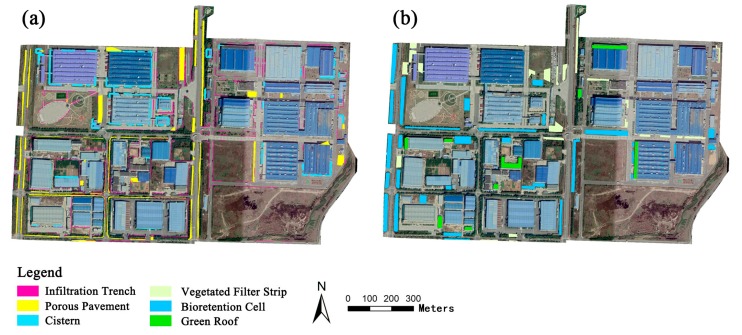
Suitability maps for gray and green stormwater control measures (SCMs). (**a**) Suitability maps for gray SCMs. (**b**) Suitability maps for green SCMs.

**Figure 4 ijerph-17-00998-f004:**
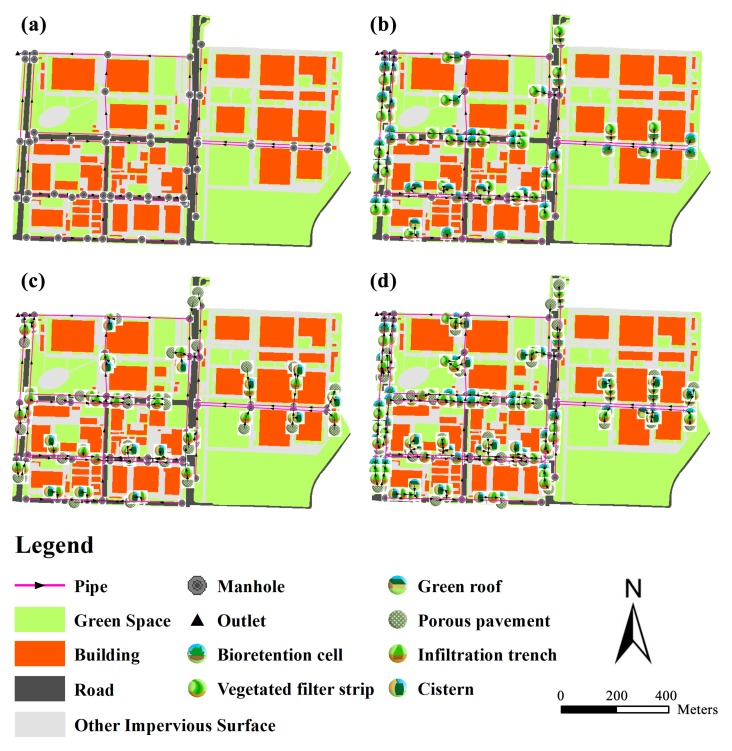
Status quo and SCM scenarios. (**a**) S0: Status quo scenario. (**b**) S1: Green SCMs only scenario. (**c**) S2: Gray SCMs only scenario. (**d**) S3: Coupled green–gray SCMs scenario.

**Figure 5 ijerph-17-00998-f005:**
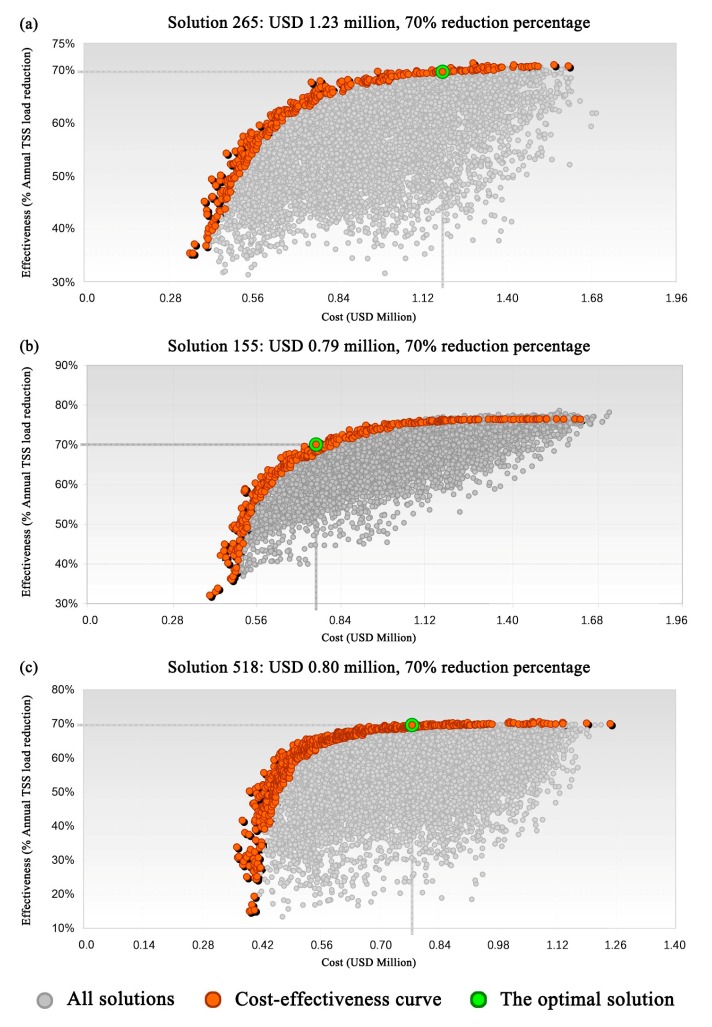
Cost-effectiveness curves and optimal solutions for different scenarios. (**a**) S1. (**b**) S2. (**c**) S3.

**Figure 6 ijerph-17-00998-f006:**
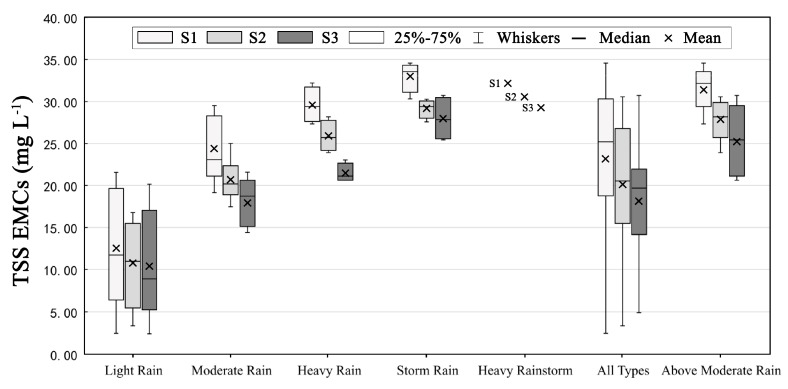
Box plot of event mean concentrations (EMCs) under different rainfall event types.

**Figure 7 ijerph-17-00998-f007:**
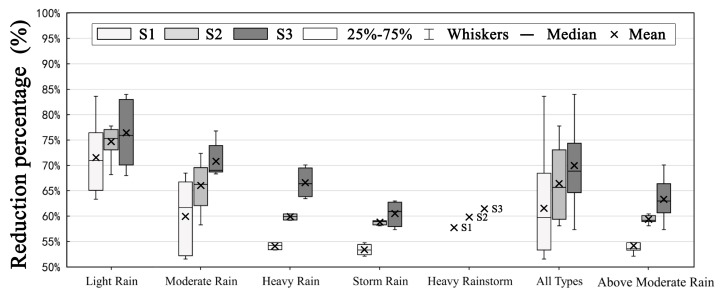
Box plot of pollution reduction percentage under different rainfall event types.

**Table 1 ijerph-17-00998-t001:** Land use distribution in the study area.

Land Use Type	Green Space	Building	Road	Other Impervious Surface
Area (ha)	51.46	39.47	35.90	12.07
Percentage (%)	37.05	28.42	25.84	8.69

**Table 2 ijerph-17-00998-t002:** Dimensions and cost of different stormwater control measures (SCMs).

SCMs	Structural Parameters					Soil Parameters		Cost (USD)
	Length (m)	Width (m)	Height (m)	Surface Storage (m)	Bottom Storage (m)	Depth (m)	Void Ratio	
IT ^1^	Depends on location	1.8	--	0.15	--	0.2	0.6	38.15/m^3^
PP ^2^	Depends on location	5	--	--	--	0.24	--	28.84/m^2^
CS ^3^	6	--	4.8	--	--	--	--	55.80/m^2^
GR ^4^	Depends on location		--	--	--	0.3	0.4	46.5/m^2^
BC ^5^	Depends on location		--	0.2	0.3	1	0.6	121.38/m^2^
VFS ^6^	Depends on location	10	Side Slope	Surface Slope	Roughness	0.2	0.6	65.98/m^2^
			1:3	4%	0.2			

^1^ Infiltration trench; ^2^ Porous pavement; ^3^ Cistern; ^4^ Green roof; ^5^ Bioretention cell; ^6^ Vegetated filter strip.

**Table 3 ijerph-17-00998-t003:** Parameters for the runoff simulation.

Parameter	Input Value	Parameter	Input Value
N-Imperv ^1^	0.005	Max.Infil.Rate ^4^ (mm/h)	76.2
N-Perv ^2^	0.15	Min.Infil.Rate ^5^ (mm/h)	10.16
Conduits’ Manning’s N	0.04	Decay Constant	5
DStore-Imperv ^3^ (mm)	0.1	DStore-Perv ^6^ (mm)	2.54

^1^ Manning’s N for impervious area; ^2^ Manning’s N for pervious area; ^3^ Depth of depression storage on impervious area; ^4^ Maximum infiltration rate; ^5^ Minimum infiltration rate; ^6^ Depth of depression storage on pervious area.

**Table 4 ijerph-17-00998-t004:** Contamination accumulation and erosion parameters for the water quality simulation.

Land Use Type	Buildup Parameters		Wash-off Parameters	
Building	Max. Buildup (kg/ha)	123.8	Coefficient	0.007
	Rate Constant (d)	0.31	Exponent	1.8
Green Space	Max. Buildup (kg/ha)	105.6	Coefficient	0.004
	Rate Constant (d)	0.9	Exponent	1.2
Road/Other Impervious Surface	Max. Buildup (kg/ha)	114.66	Coefficient	0.008
	Rate Constant (d)	1.02	Exponent	1.8

**Table 5 ijerph-17-00998-t005:** Areas and proportions of different SCMs.

SCMs	IT	PP	CS	GR	BC	VFS	Total
Area (ha)	4.62	4.45	2.61	1.16	5.34	4.04	16.14
Proportion (%)	3.33	3.21	1.88	0.84	3.85	2.91	16.02

**Table 6 ijerph-17-00998-t006:** Composition of optimal solutions under different scenarios.

Scenarios	SCMs	Cost Proportion (%)	Total Cost (USD Million)	Total Area (ha)
S1	BC	52	0.64	0.53
	GR	1	0.01	0.03
	VFS	47	0.58	0.87
	Total	100	1.23	1.43
S2	CS	6	0.05	0.06
	PP	60	0.48	1.65
	IT	34	0.26	3.53
	Total	100	0.79	5.24
S3	CS	27	0.22	0.12
	BC	20	0.16	0.13
	PP	6	0.05	0.17
	GR	3	0.02	0.05
	VFS	23	0.18	0.28
	IT	21	0.17	2.19
	Total	100	0.80	2.94

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
