# Peer review of "Cost-Effectiveness Analysis of Green–Gray Stormwater Control Measures for Non-Point Source Pollution"

_ijerph, 2020, doi:10.3390/ijerph17030998_

Round 1
Reviewer 1 Report
Dear Authors,
Your submitted manuscript has an actual and of increasing interest subject: urban stormwater control.
In this manuscript, a lot of various scientific fields (i.e. scientific/technical, economic, environmental, social ones) are treated in an interconnected manner by using three scenarios of selection different stormwater control measures (i.e. green, grey and coupled green-grey SCMs) related to a reference existing one (current post-development situation without any new SCMs), and a selection methodology based on cost-effectiveness criteria referring to a few hydrological and hydraulic operating parameters, especially outflow and total suspended solids (TSS) concentration, the optimization methodology based on SUSTAIN model developed by U.S.A. (a hydrological model), BMP module and NSSA-II algorithm.
I like your manuscript which can improve by considering my suggestions or recommendations.
Specific comments and recommendations:
Few corrections in the Abstract are necessary; e.g., (i) lines 14 - it is present a sentence without any sense, meaning without subjective and verb, etc.; (ii) cost must express in international equivalent monetary units (i.e. $ not yuan); (ii) English correction is imposed, especially at line 13. Few gramary and English corrections in Introduction section are necessary, e.g., line 63, line 75-78 i.e. (1) creation..., (2) comparison ...(3) investigation... In the manuscript text must write completely the date, meaning September not Sept. At section 2.2., the terms as 'dominant parameters calibration..', 'deterministic coefficient' must be avoided; can replace with 'correlation coefficient', 'correlation of significant operating parameters...' At section 2.4, in table 2, all costs must express in $ (for value validation) as a current exchange monetary unit. In table 3, no agreed abbreviation must found, i.e. instead 'Infil.' must be written Ínfiltration' line 187, correction of text as '...was placed alongside walks and...low traffic circulations.' In table 5 must maintain the same order of SCMs as in table 2 (IT, PP, CS, GR, BC, VFS) In figure 4, instead 'conduit' can use the term of 'pipe', and at Conclusions section, 'yuan' must express in equivalent as $. At References, some corrections are necessary, e.g. (i) all authors inserted at references 10, 13, 39 (it is not polite for the other authors); (ii) reference 14, must insert article doi; (iii) must verify the same writing format of all references, e.g. reference 35, etc.These are a few of my comments for manuscript authors.
Author Response
Manuscript ID: ijerph-697147
Title: Cost-effectiveness Analysis of Green-Gray Stormwater Control Measures for Non-Point Source Pollution
Section: Environmental Science and Engineering
Response to the reviewers
Reviewer #1:
We want to thank you for your constructive comments that helped us to improve the manuscript. In this response, we explain how we have addressed your questions. We follow the order of the comments given, with our responses provided in blue bold font.
Reviewers' comments:
Your submitted manuscript has an actual and of increasing interest subject: urban stormwater control. In this manuscript, a lot of various scientific fields (i.e. scientific/technical, economic, environmental, social ones) are treated in an interconnected manner by using three scenarios of selection different stormwater control measures (i.e. green, grey and coupled green-grey SCMs) related to a reference existing one (current post-development situation without any new SCMs), and a selection methodology based on cost-effectiveness criteria referring to a few hydrological and hydraulic operating parameters, especially outflow and total suspended solids (TSS) concentration, the optimization methodology based on SUSTAIN model developed by U.S.A. (a hydrological model), BMP module and NGSA-II algorithm.
I like your manuscript which can improve by considering my suggestions or recommendations.
General comments include:
Cost must express in international equivalent monetary units (i.e. $ not yuan).Reply: Suggestion was adopted. All the cost data in the manuscript have been converted into U.S. Dollar according to the exchange rate on January 29, 2020 (1 CNY=0.144018 USD). Moreover, the unit “Yuan” has been revised to “USD” in Table 2, Figure 5 and Table 6. Please see the file "Revised manuscript with changes marked".
In the manuscript text must write completely the date, meaning September not Sept.Reply: Suggestion was adopted. The phrase “Sep 19, 2018” has been revised to “September 19, 2018” as suggested. The date in reference 38 has been revised to “November 30, 2019”.
All references must be verified in the same writing format.Reply: Suggestion was adopted. All references have been verified in the same writing format. Please see the file "Revised manuscript with changes marked".
Specific comments include:
Line 14: it is present a sentence without any sense, meaning without subjective and verb, etc.Reply: Suggestion was adopted. The sentence has been revised. Please see the file "Revised manuscript with changes marked".
Line 13: English correction is imposed.Reply: Suggestion was adopted. The word “suffers” has been revised to “suffer”.
Line 63 and Line 75-78: Few grammar and English corrections in Introduction section are necessary.Reply: Suggestion was adopted. The sentence has been revised. Please see the file "Revised manuscript with changes marked".
Section 2.2: the terms as 'dominant parameters calibration..', 'deterministic coefficient' must be avoided; can replace with 'correlation coefficient', 'correlation of significant operating parameters...'.Reply: Suggestion was partly adopted. The phrase “dominant parameters calibration” has been revised to “The calibration of significant operating parameters”, and the phrase “deterministic coefficient” has been revised to “correlation coefficient”.
Table 2: all costs must express in $ (for value validation) as a current exchange monetary unit.Reply: Suggestion was adopted. The unit of costs have been converted into US Dollar according to the exchange rate (1 CNY= 0.144018 USD) on January 29, 2020.
Table 3: no agreed abbreviation must found, i.e. instead 'Infil.' must be written Infiltration'.Reply: Suggestion was adopted. Superscript description and a separate glossary list to explain all the abbreviations have been added in/below the table.
Line 187: correction of text as '...was placed alongside walks and...low traffic circulations.'.Reply: Suggestion was not adopted. We think the meaning of this sentence is correct.
Table 5: the order of SCMs must maintain the same as in Table 2.Reply: Suggestion was adopted. The order of SCMs in Table 5 has been revised to be the same as in the Table 2.
Figure 4: instead 'conduit', use the term of 'pipe'.Reply: Suggestion was adopted. The word “conduit” in Figure 4 has been revised to “pipe”.
Reference 10, 13 and 39: all authors should be inserted.Reply: Suggestion was adopted. All authors have been inserted in reference 10, 13 and 39. Please see the file "Revised manuscript with changes marked".
Reference 14: article doi should be inserted.Reply: Suggestion was adopted. Article doi has been inserted in reference 14.
Reviewer 2 Report
Very interesting paper. The way they defined "Gray" is very interesting. In USA, we define "Gray" as pipes or storage tanks. Porous pavements, infiltration trench, etc. are defined as "Green".
Thoroughly enjoyed the paper. Did not come across any English language problems.
Author Response
Manuscript ID: ijerph-697147
Title: Cost-effectiveness Analysis of Green-Gray Stormwater Control Measures for Non-Point Source Pollution
Section: Environmental Science and Engineering
Response to the reviewers
Reviewer #2:
We want to thank you for your constructive comments that helped us to improve the manuscript. In this response, we explain how we have addressed your questions. We follow the order of the comments given, with our responses provided in blue bold font.
Reviewers' comments:
Very interesting paper. The way they defined "Gray" is very interesting. In USA, we define "Gray" as pipes or storage tanks. Porous pavements, infiltration trench, etc. are defined as "Green". Thoroughly enjoyed the paper. Did not come across any English language problems.
Reply: Most of previous studies classified stormwater control measures into green and gray infrastructure, based on their geometry, construction material and contributing area. While, there are also studies indicating that SCMs with similar form and function can be assigned different classification across 9 cities in USA and regard different SCMs as a continuum (Bell et al., 2019). They suggested that grayer SCMs are typically constructed of traditional, coarser-grained materials and rarely include soil or plant materials, while greener SCMs are filled with both coarse-grade, fine-grade media and soil to support the growth of plants in them. Considering that a specific SCM can shift classifications by changing a few design variable or location based on the discrete classification scheme, we adopt their definitions for the division of gray or green SCMs in this study.

Reviewer 3 Report
Lack consequences please compare lines 19, 20 and lines 216, 217
Figure 6 X-axis unclear descrition
Figure 7 X-axis change description
Author Response
Manuscript ID: ijerph-697147
Title: Cost-effectiveness Analysis of Green-Gray Stormwater Control Measures for Non-Point Source Pollution
Section: Environmental Science and Engineering
Response to the reviewers
Reviewer #3:
We want to thank you for your constructive comments that helped us to improve the manuscript. In this response, we explain how we have addressed your questions. We follow the order of the comments given, with our responses provided in blue bold font.
Reviewers' comments:
General comments:
Lack consequences please compare lines 19, 20 and lines 216, 217.Reply: We have compared the consequences in line 19-20 and line 216-217. The optimal solutions’ costs for three scenarios are CNY 8.50 million, CNY 5.50 million and CNY 5.55 million, respectively. According to another reviewer’ comment, we have converted the monetary unit from CNY to USD based on the exchange rate on January 29, 2020 in the revision process (1 CNY=0.144018 USD). Therefore, the optimal solution’s costs of S1, S2 and S3 are USD 1.23, 0.79, and 0.80 million, respectively.
Specific comments:
Figure 6 and 7: X-axis’ description is unclear.Reply: Suggestion was adopted. We have redrawn Figure 6 and Figure 7 from curve diagram to box plots. The data tags on X-axis have been revised to “light rain”, “moderate rain”, “heavy rain”, “storm rain”, “heavy rainstorm”, “all types” and “above moderate rain”, representing the different rainfall event types.

Reviewer 4 Report
In the manuscript ID: ijerph-697147, entitled: “Cost-effectiveness Analysis of Green-Gray Stormwater Control Measures for Non-Point Source Pollution”, the authors compare the cost-effectiveness of 3 different stormwater control scenarios. The manuscript is well written and of interest, although I suggest to emphasize the importance of the study. In my opinion the manuscript is suitable for publication and I suggest some minor revisions:
Line 40: Can authors better define “sponge cities”?
Line 85: Can authors better define “other impervious surface”?
Equation 1: I suggest eliminating “and”.
Table 5: I suggest adding a superscript description, also in this case.
Figure 4: In my opinion, these figures should be redone (the legend is clear, but not visible in the different scenarios).
Author Response
Manuscript ID: ijerph-697147
Title: Cost-effectiveness Analysis of Green-Gray Stormwater Control Measures for Non-Point Source Pollution
Section: Environmental Science and Engineering
Response to the reviewers
Reviewer #4:
We want to thank you for your constructive comments that helped us to improve the manuscript. In this response, we explain how we have addressed your questions. We follow the order of the comments given, with our responses provided in blue bold font.
Reviewers' comments:
In the manuscript ID: ijerph-697147, entitled: “Cost-effectiveness Analysis of Green-Gray Stormwater Control Measures for Non-Point Source Pollution”, the authors compare the cost-effectiveness of 3 different stormwater control scenarios. The manuscript is well written and of interest, although I suggest to emphasize the importance of the study. In my opinion the manuscript is suitable for publication and I suggest some minor revisions.
General comments:
Line 40: Can authors better define “sponge cities”?Reply: “Sponge City” is a new terminology surrounding urban drainage, announced by Chinese government in 2013. It implies that the city has good flexibility in adapting to environmental changes and responding to natural disasters caused by precipitation like a sponge. Sponge city construction comprises aspects of urban drainage, urban planning, landscape and more (Che and Zhang, 2019), aiming at runoff total volume control, runoff peak control, runoff pollution control and rainwater harvesting. Functional elements in sponge cities compromise sponge infrastructures, decentralized sewage system, fit-for-purpose water supply system and near-natural ecological zones (Ren et al., 2017).
Line 85: Can authors better define “other impervious surface”?
Reply: “Other impervious surface” in the study area indicates inner roads with low traffic load, parking lots and footpaths within factory area. This kind of impervious surfaces are mostly covered by cement concrete instead of asphalt on roads.
